# Detection of transient synchrony across oscillating receptors by the central electrosensory system of mormyrid fish

Alejandro Vélez*, Bruce A Carlson*

Department of Biology, Washington University in St. Louis, St. Louis, United States

**Abstract** Recently, we reported evidence for a novel mechanism of peripheral sensory coding based on oscillatory synchrony. Spontaneously oscillating electroreceptors in weakly electric fish (Mormyridae) respond to electrosensory stimuli with a phase reset that results in transient synchrony across the receptor population (*Baker et al., 2015*). Here, we asked whether the central electrosensory system actually detects the occurrence of synchronous oscillations among receptors. We found that electrosensory stimulation elicited evoked potentials in the midbrain exterolateral nucleus at a short latency following receptor synchronization. Frequency tuning in the midbrain resembled peripheral frequency tuning, which matches the intrinsic oscillation frequencies of the receptors. These frequencies are lower than those in individual conspecific signals, and instead match those found in collective signals produced by groups of conspecifics. Our results provide further support for a novel mechanism for sensory coding based on the detection of oscillatory synchrony among peripheral receptors.

*For correspondence:
avelezmelendez@wustl.edu (AV);
carlson.bruce@wustl.edu (BAC)

**Competing interests:** The authors declare that no competing interests exist.

## Introduction

We recently proposed a novel mechanism for peripheral sensory coding based on the detection of transient synchrony among oscillatory receptors in weakly electric fish of the family Mormyridae (*Baker et al., 2015*). Whether the central electrosensory system actually detects the occurrence of synchronous receptor oscillations remained unknown. Here, we tested this hypothesis using in vivo evoked potential recordings from a midbrain nucleus in the ascending electrosensory pathway of mormyrids.

Mormyrids communicate using pulse-type electric organ discharges (EODs) that are highly stereotyped and species-specific (*Hopkins, 1981*). The ability to detect subtle variations in EOD waveform evolved independently in two mormyrid lineages (*Carlson et al., 2011*). One lineage belongs to the subfamily Mormyrinae, is known as 'clade A', and includes over 175 species (*Carlson and Arnegard, 2011*). The other lineage has only one known extant species, *Petrocephalus microphthalmus*, and belongs to the subfamily Petrocephalinae. All other petrocephaline species studied so far are unable to detect subtle variations in EOD waveform (*Carlson et al., 2011*). Interestingly, the perceptual ability to detect signal variation is associated with parallel evolutionary changes in the peripheral and central electrosensory system (*Baker et al., 2015*; *Carlson et al., 2011*).

Electric communication signals are detected by electroreceptors called knollenorgans. In species sensitive to EOD waveform variation (i.e., all clade A species and *P. microphthalmus*), knollenorgans are distributed throughout the surface of the head, back, and belly (*Baker et al., 2015*; *Carlson et al., 2011*). These receptors fire spontaneous spikes and respond to EODs with a single time-locked spike (*Baker et al., 2015*). In contrast, knollenorgans are limited to three rosettes on each side of the head in species that are unable to detect variation in EOD waveform (*Baker et al., 2015*; *Carlson et al., 2011*). Receptors in species unable to detect EOD waveform variation produce

spontaneously oscillating potentials at frequencies between 1 and 3 kHz. Spontaneous oscillations across these receptors are uncorrelated (*Baker et al., 2015*). Interestingly, however, these oscillating receptors respond to electrosensory stimulation with an increase in the amplitude of oscillations and with a phase reset that causes the oscillations across receptors to transiently synchronize (*Baker et al., 2015*). Here, we build upon this finding by asking whether this transient oscillatory synchrony across receptors is detected by the central electrosensory system.

The ability to detect EOD waveform variation is also associated with parallel changes in the anatomy of the central electrosensory system (*Carlson et al., 2011*; *Figure 1A–C*). In species unable to detect EOD waveform variation, the midbrain exterolateral nucleus (EL) is small and undifferentiated. In clade A and *P. microphthalmus*, however, the EL is enlarged and divided into separate anterior (ELa) and posterior (ELp) regions. Our current understanding of signal processing in EL is limited to studies on a few clade-A species (reviewed in *Baker et al., 2013*). Briefly, spiking receptors on the surface of the skin project to the ipsilateral nucleus of the electrosensory lateral line lobe (nELL) in the hindbrain (*Bell and Grant, 1989*). The nELL relays electrosensory information bilaterally to ELa. The axons of nELL cells synapse onto two types of cells in ELa: small cells and large cells. Small cells in ELa perform the first stages of EOD waveform analysis in a circuit that includes direct inhibition from large cells and excitation from nELL cells with axonal projections that vary in length. These axonal projections serve as delay lines and establish differences in the relative timing between inhibitory and excitatory input, allowing small cells to process submillisecond time differences based on an anti-coincidence detection mechanism (*Friedman and Hopkins, 1998*; *Lyons-Warren et al., 2013a*). Small cells in ELa then project to ELp, in which multipolar cells perform the first stages of inter-pulse interval analysis (*Carlson, 2009*). How the central electrosensory systems of *P. microphthalmus* and species with a small and undifferentiated EL process communication signals is still an open question.

We used in vivo evoked potential recordings from the EL to test the hypothesis that the central electrosensory system detects transient synchrony among oscillating peripheral receptors. We show here, for the first time, midbrain responses to electrosensory stimulation in a species with oscillating receptors and a small, undifferentiated EL (*Petrocephalus tenuicauda*). We also document midbrain electrosensory responses in *P. microphthalmus* and *Brevimyrus niger*, two species in which spiking receptors and an enlarged, subdivided ELa/ELp evolved independently.

## Results

### Electrosensory stimulation elicited time-locked evoked potentials in the EL

Electrosensory stimulation with conspecific EODs elicited time-locked evoked potentials in the EL of *P. tenuicauda* (*Figure 1D*, n = 8). The latency to the negative peak ranged between 2.56 and 7.26 ms following stimulus onset (mean ± sd = 4.29 ± 1.57 ms). The sharpness of the evoked potentials, measured as the relative latency between the maximum and minimum values of the evoked potential, varied between 0.62 and 2.98 ms (mean ± sd = 1.41 ± 0.82 ms). Latency and sharpness varied depending on the location of the recording electrode along an anterior-posterior axis of EL, with longer latencies and broader evoked potentials occurring towards the posterior end.

We also obtained evoked potentials elicited by conspecific EODs in three *P. microphthalmus* and two *B. niger* (*Figure 1E,F*). In *P. microphthalmus*, evoked potentials in ELa were sharp (range = 0.43–0.60 ms, mean ± sd = 0.54 ± 0.10 ms) and had short latencies (range = 2.65–2.90 ms, mean ± sd = 2.80 ± 0.14 ms), while evoked potentials in ELp were broader (range = 3.09–3.19 ms, mean ± sd = 3.15 ± 0.05 ms) and had longer latencies (range = 7.22–7.58 ms, mean ± sd = 7.44 ± 0.19 ms). Similarly, evoked potentials in *B. niger* were sharp (range = 0.66–0.74 ms, mean ± sd = 0.70 ± 0.06 ms) and had short latencies (range = 3.73–3.81 ms, mean ± sd = 3.77 ± 0.06 ms) in ELa, and were broader (range = 2.83–3.85 ms, mean ± sd = 3.34 ± 0.72 ms) with longer latencies (range = 7.63–8.83 ms, mean ± sd = 8.23 ± 0.85 ms) in ELp.

### Evoked potentials in EL are blocked by a corollary discharge of the EOD motor command

Previous studies on clade-A species have shown that responses in the knollenorgan electrosensory system are blocked shortly after the fish produces an EOD (*Amagai, 1998*; *Bell and Grant, 1989*;

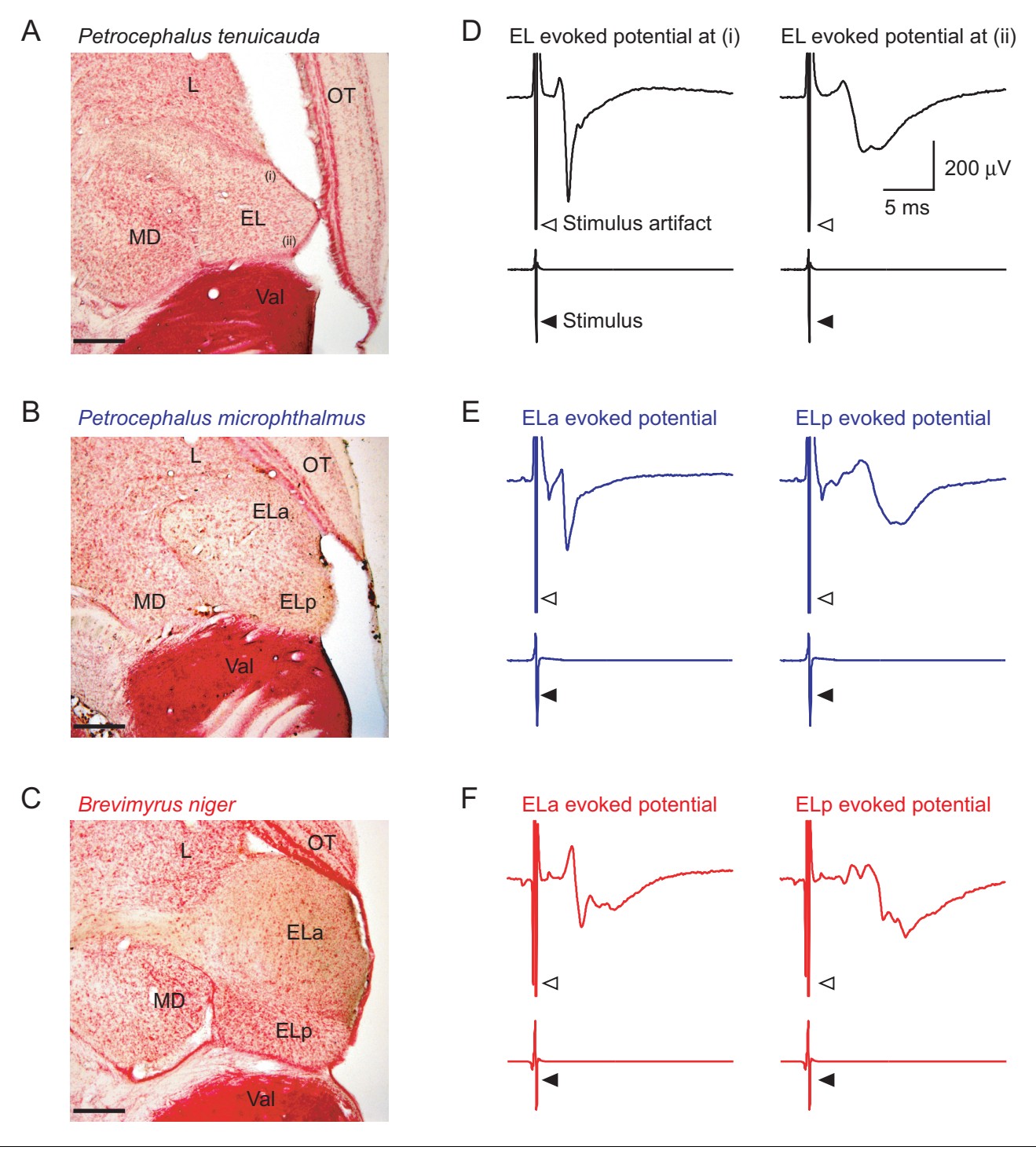

**Figure 1.** The central electrosensory system detects transient synchrony among oscillating receptors. 50 μm horizontal sections of the midbrain of *P. tenuicauda* (A), *P. microphthalmus* (B), and *B. niger* (C). The midbrain exterolateral nucleus (EL) in *P. tenuicauda* is small and undifferentiated; in the other two species, EL is enlarged and subdivided into separate anterior (ELa) and posterior (ELp) nuclei. (D) Representative mean evoked potentials (n = 10 traces) from the EL of *P. tenuicauda* obtained from relatively more anterior (left) and posterior (right) regions. Representative mean evoked potentials (n = 10 traces) obtained from ELa (left) and ELp (right) in *P. microphthalmus* (E) and *B. niger* (F). Scale bars in (A), (B), and (C) represent 200 μm. L, lateral nucleus; OT, optic tectum; MD, mediodorsal nucleus; Val, valvula cerebellum.

*Bennett and Steinbach, 1969*; *Russell and Bell, 1978*). Every time an EOD command is generated, corollary discharge inhibition blocks electrosensory responses in the hindbrain nELL. Therefore, the ELa and ELp in clade-A species do not respond to the fish's own EOD and are devoted to processing communication signals produced by other individuals. We asked whether electrosensory responses are also blocked by corollary discharge inhibition in the knollenorgan pathways of *P. tenuicauda* and *P. microphthalmus*.

We recorded evoked potentials in response to 0.5 ms bipolar square electric pulses that were delivered with a delay between 0 and 6 ms (in 0.5 ms steps) with respect to the spinal electromotor neuron (EMN) command volley. The EMN command volley persists in paralyzed and awake fish, although EOD production is blocked. Evoked potentials in the EL of *P. tenuicauda* were attenuated when the stimulus was presented with a delay of 2.5–3.5 ms (*Figure 2A*; ANOVA: $F_{12,84} = 19.31$, $p<0.0001$, $\eta^2 = 0.73$). Evoked potentials were also attenuated over the same range in the ELa and ELp of both *P. microphthalmus* (*Figure 2B*; ELa: $F_{12,60} = 30.8$, $p<0.0001$, $\eta^2 = 0.86$; ELp: $F_{12,60} = 40.52$, $p<0.0001$, $\eta^2 = 0.89$) and *B. niger* (*Figure 2C*; ELa: $F_{12,60} = 8.97$, $p<0.0001$, $\eta^2 = 0.64$; ELp: $F_{12,60} = 7.46$, $p<0.0001$, $\eta^2 = 0.60$).

## Midbrain frequency tuning reflects the intrinsic oscillation frequencies of oscillating receptors

We generated frequency tuning curves for the midbrain EL in *P. tenuicauda*, and for both ELa and ELp in *P. microphthalmus* and *B. niger*. These curves spanned the range of frequencies present in conspecific natural EODs of all three species (*Figure 3*). We measured evoked potentials in response to bipolar single-cycle sinusoidal electric pulses varying in frequency (0.17–27.9 kHz, *Figure 4A*), at three different intensities (73.6, 23.4, and 7.4 mV/cm). The amplitudes of evoked potentials in the EL of *P. tenuicauda* varied with frequency ($F_{10,177} = 79.08$, $p<0.0001$, $\eta^2 = 0.73$), and were highest at frequencies between 0.5 and 3 kHz (*Figure 4B*). Frequency tuning in EL matched the peripheral frequency tuning previously described (*Baker et al. 2015*). Further, the range of frequencies of best sensitivity was much lower than the dominant frequencies present in conspecific EODs (~9 kHz) (*Baker et al., 2015*; *Figure 3A*, *4B*). The amplitude of evoked potentials also decreased with stimulus intensity ($F_{2,177} = 58.96$, $p<0.0001$, $\eta^2 = 0.11$).

In species with spiking receptors, the amplitude of evoked potentials also depended on stimulus frequency (*P. microphthalmus* ELa: $F_{10,84} = 25.26$, $p<0.0001$, $\eta^2 = 0.46$; ELp: $F_{10,84} = 32.85$, $p<0.0001$, $\eta^2 = 0.41$; *B. niger* ELa: $F_{10,84} = 27.39$, $p<0.0001$, $\eta^2 = 0.44$; ELp: $F_{10,84} = 17.18$, $p<0.0001$, $\eta^2 = 0.42$; *Figure 4C–F*). In both species, the shape of the tuning curve was similar between ELa and ELp and had peak sensitivity at frequencies between 2 and 5 kHz (*Figure 4C–F*). Midbrain frequency tuning was very similar to that of the spiking receptors and more closely matched the frequency content of conspecific EODs in both species (*Baker et al., 2015*; *Figure 3A*, *4C–F*). The amplitude of evoked potentials also decreased with stimulus intensity (*P. microphthalmus* ELa: $F_{2,84} = 106.39$, $p<0.0001$, $\eta^2 = 0.39$; ELp: $F_{2,84} = 149.59$, $p<0.0001$, $\eta^2 = 0.42$; *B. niger* ELa: $F_{2,84} = 133.48$, $p<0.0001$, $\eta^2 = 0.43$; ELp: $F_{2,84} = 75.48$, $p<0.0001$, $\eta^2 = 0.37$).

## Discussion

Our results provide further support for a novel mechanism for sensory coding based on the detection of oscillatory synchrony among sensory receptors. We show here that electrosensory stimuli that cause transient oscillatory synchrony among receptors elicit time-locked evoked potentials in the midbrain of mormyrid species with oscillating receptors. We also show that, just like the ELa/ELp of clade-A species, the EL of *P. tenuicauda* and ELa/ELp of *P. microphthalmus* is a brain region devoted to processing communication signals produced by other individuals: evoked potentials are blocked during a short window soon after the fish produces an EOD command. In addition, we show that peripheral frequency tuning is maintained in the midbrain and, in species with oscillating receptors, frequency sensitivity matches the intrinsic oscillation frequencies of the receptors.

In species with oscillating receptors, electrosensory stimuli elicit an increase in the amplitude of oscillations and a phase reset that results in transient oscillatory synchrony across receptors (*Baker et al., 2015*). To conclusively demonstrate that the central electrosensory system detects this transient synchrony, an experiment manipulating only the degree of synchrony across the population of receptors, independent of the stimulus, is necessary. However, there is evidence to suggest that

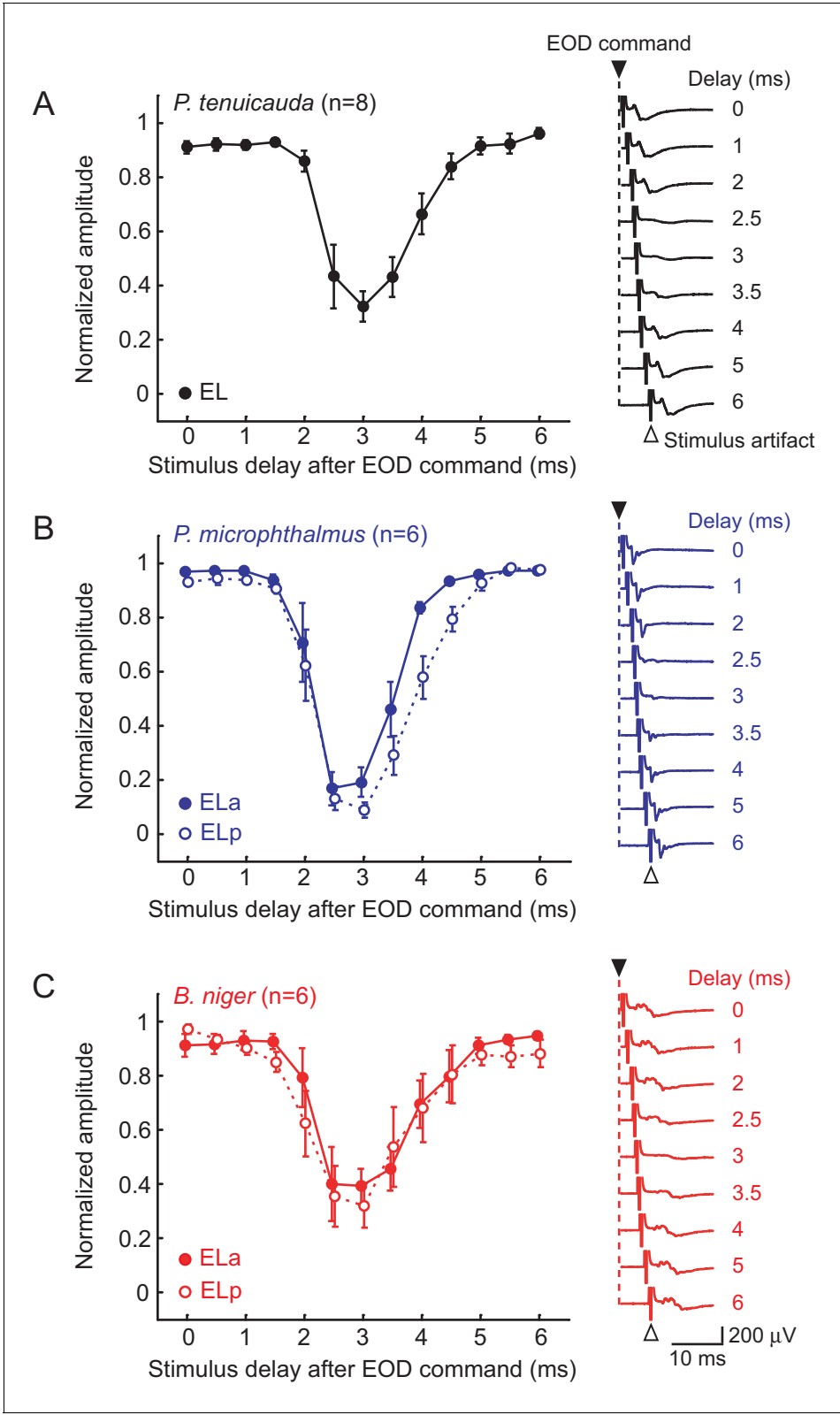

**Figure 2.** Evoked potentials in the midbrain are blocked by a corollary discharge of the EOD motor command. Effect of stimulus delay with respect to the spinal EOD motor command on the amplitude of evoked potentials in *P. tenuicauda* (A), *P. microphthalmus* (B) and *B. niger* (C). Evoked potentials in *P. tenuicauda* were obtained throughout EL, in locations varying from more anterior to more posterior, as in (i) and (ii) in **Figure 1A**. In (B) and (C), full symbols represent data from the anterior division of the exterolateral nucleus (ELa) and open symbols represent data from the posterior division (ELp).
*Figure 2 continued on next page*

*Figure 2 continued*

We measured the peak-to-peak amplitude of the evoked potential at each latency and normalized them to the maximum value for each fish tested and at each nucleus. Traces on the right are representative mean evoked potentials (n = 10 traces) when the stimulus was delivered at delays of 0, 1, 2, 2.5, 3, 3.5, 4, 5, and 6 ms with respect to the EOD command. Each symbol in (A), (B), and (C) represent the mean normalized amplitude and the error bars represent the S.E.M.

the electrosensory system indeed detects transient oscillatory synchrony. Oscillatory receptors synchronize at latencies of ~0.2 ms following stimulus onset and rapidly desynchronize due to differences across receptors in their intrinsic oscillation frequencies (*Baker et al., 2015*). The relative latencies between receptor synchronization and midbrain evoked potentials is comparable to the relative latencies between receptor spiking and midbrain evoked potentials previously reported in species with spiking receptors (*Amagai, 1998*; *Amagai et al., 1998*; *Friedman and Hopkins, 1998*; *Lyons-Warren et al., 2013a*). In fact, the range of latencies of evoked potentials in the EL of *P. tenuicauda* spans the range of latencies recorded in the ELa and ELp of both *P. microphthalmus* and *B. niger*. Further, the amplitudes of both spontaneous and stimulus-evoked oscillations vary widely across the population of receptors, whereas the phase reset is highly consistent for sufficiently strong stimuli (*Baker et al., 2015*). Thus, synchrony across receptors resulting from stimulus-induced phase resets is the only known peripheral event that reliably, and exclusively, precedes the evoked potential responses we observed in EL.

We hypothesize that detection of transient oscillatory synchrony by the central electrosensory system is likely based on a coincidence detection mechanism. Such coincidence detection would have to take place before EL, in either of two possible locations along the ascending electrosensory pathway: the primary afferents or the nELL of the hindbrain. Primary afferents could detect synchrony if they innervate several knollenorgans, whereas nELL cells could detect synchrony if they receive input from multiple afferents. Previous studies have shown that individual afferent fibers do not project to multiple knollenorgans in species with oscillating receptors (*Harder, 1968*). Further, nELL neurons in clade-A species with spiking receptors have anatomical features typical of coincidence detectors (*Ashida et al., 2007*; *Bell et al., 1981*; *Kuba et al., 2006*; *Mugnaini and Maler, 1987*; *Szabo and Ravaille, 1976*), and they receive convergent input from multiple primary afferents (*Bell and Grant,*

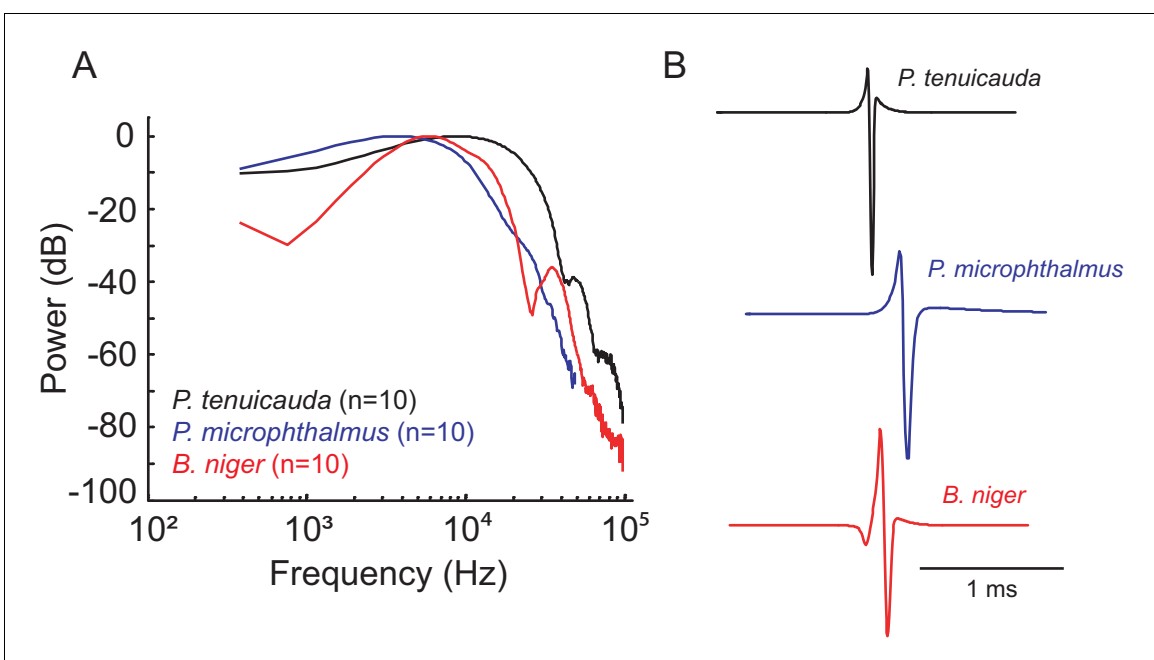

**Figure 3.** Frequency content relates to the duration of communication signals in pulse-type mormyrids. (A) Power spectra and (B) representative waveforms of electric organ discharges (EODs) of the three species tested.

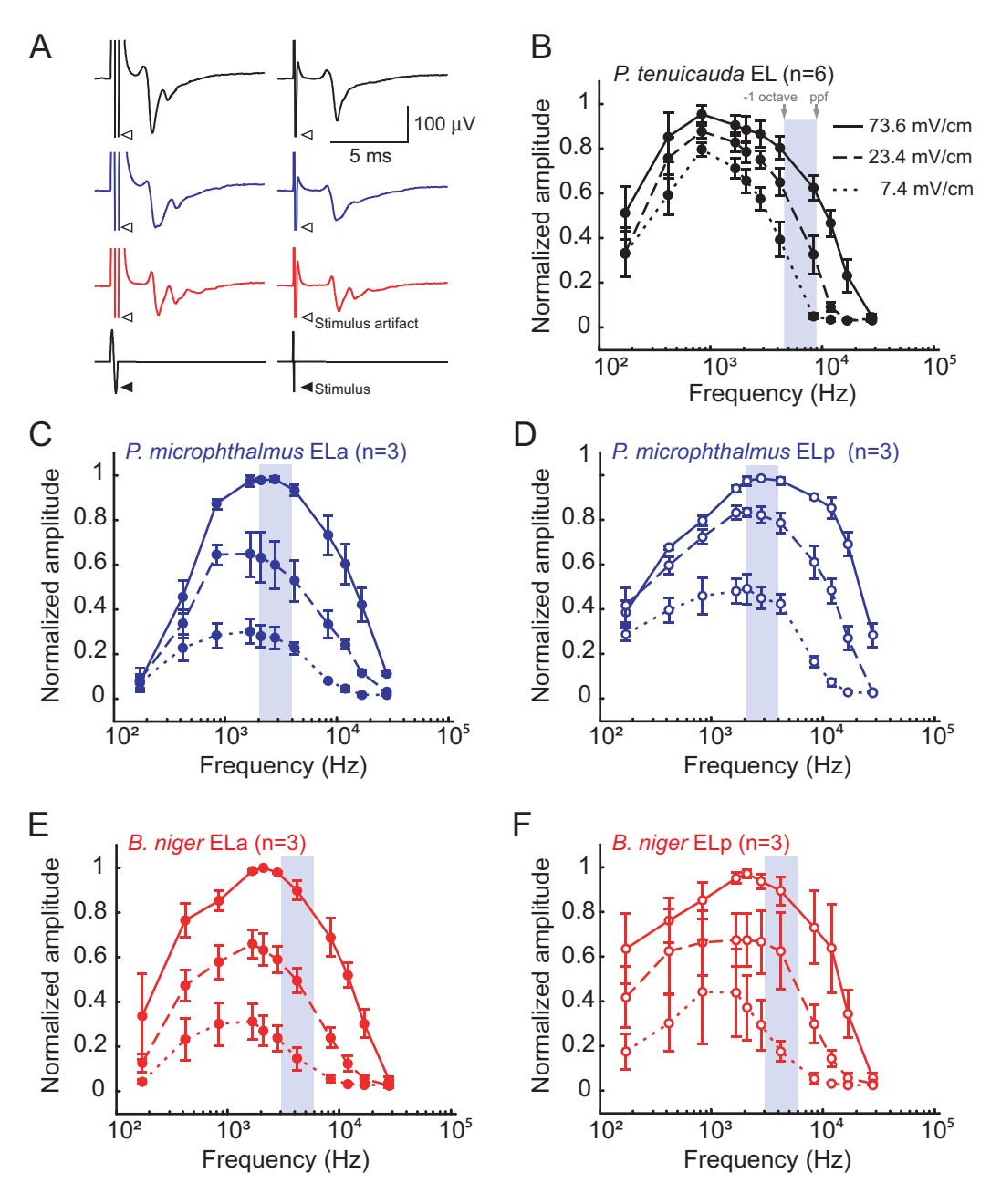

**Figure 4.** Midbrain frequency sensitivity matches EOD frequency content in species with spiking receptors, but not in species with oscillating receptors. (**A**) Representative mean evoked potentials (n = 10 traces) from the EL of *P. tenuicauda* (black), ELa of *P. microphthalmus* (blue), and ELa of B. *niger* (red) in response to single-cycle bipolar sinusoidal pulses with durations of 0.5ms (left) and 0.1 ms (right), which correspond to peak power frequencies of 1.7 and 8.4 kHz, respectively. Frequency tuning curves of the midbrain exterolateral nucleus (EL) in *P. tenuicauda* (**B**), and the anterior (ELa) and posterior (ELp) subnuclei of EL in *P. microphthalmus* (**C** and **D**, respectively) and *B. niger* (**E** and **F**, respectively). Evoked potentials in *P. tenuicauda* were obtained throughout EL, in locations varying from more anterior to more posterior, as in (i) and (ii) in *Figure 1A*. Frequency tuning curves were obtained at three intensities: 73.6 (full lines), 23.4 (dashed lines), and 7.4 (dotted lines) mV/cm. We measured the peak-to-peak amplitude of the evoked potential at each frequency and intensity, and normalized them to the maximum value for each fish and each nucleus separately. Gray boxes represent the frequencies between the peak power frequency (ppf) of the conspecific EOD and one octave below the ppf. Each symbol in (**B**), (**C**), (**D**), (**E**), and (**F**) represent the mean normalized amplitude and the error bars represent the S.E.M.

*1989*). These results suggest that coincidence detection in species with oscillatory receptors could take place in the nELL. However, detailed studies of primary afferent innervations and of the nELL are necessary to conclusively determine whether coincidence detection takes place and, if so, where.

We show that peripheral frequency tuning is maintained in the midbrain. The electrosensory system of species with spiking receptors is tuned to the frequency content of conspecific EODs (*Bass and Hopkins, 1984*; *Hopkins and Bass, 1981*; *Hopkins, 1981*; *Lyons-Warren et al., 2012*). In species with oscillating receptors, however, peripheral and central frequency sensitivity match the intrinsic oscillations of the receptors (1–3 kHz), which are much lower than the frequency content of conspecific EODs (~9 kHz) (*Baker et al., 2015*; this study). Instead, the electrosensory system of species with oscillating receptors appears to be tuned to properties of the collective signals produced by groups of conspecific individuals (*Baker et al., 2015*). Field and laboratory observations indicate that many species with oscillating receptors form shoals in open water (*Hopkins, 1980*; *Lavoué et al., 2004*). In contrast, most species with spiking receptors tend to be solitary, seek shelter, and defend territories (*Hopkins, 1980*; *Lavoué et al., 2004*). Indeed, we have recently shown that a species with spiking receptors shows social competition, whereas a sympatric species with oscillating receptors shows social affiliation, both in the field and laboratory settings (*Carlson, 2016*). Thus, differences in the physiological properties of the electrosensory system likely reflect adaptations for different social environments across mormyrids. However, *P. microphthalmus* is an interesting exception in that frequency sensitivity matches the frequency content of individual EODs, but they form shoals in the laboratory and in natural settings (*Baker et al., 2015*; *Lavoué et al., 2004*).

Interestingly, stimulus intensity had a larger effect on species with spiking receptors ($\eta^2 \geq 0.37$) than in species with oscillating receptors ($\eta^2 = 0.11$; *Figure 4*). At low intensities, evoked potentials were relatively stronger in *P. tenuicauda* than in *P. microphthalmus* and *B. niger*. These results suggest that the peripheral electrosensory system is more sensitive in species with oscillating receptors than in species with spiking receptors. The enhanced sensitivity in *P. tenuicauda* may be a property of a network of resonating oscillators: weak signals are more efficiently detected and amplified by a coherent summation of oscillators (*Buzsáki and Draguhn, 2004*; *Kandel and Buzsáki, 1997*; *Steriade and Timofeev, 2003*). Furthermore, stimulus intensity could be analyzed by the electrosensory system using a synchrony code. In *P. tenuicauda*, the small changes in the amplitude of evoked potentials as a function of stimulus intensity suggests little differences in the amount of synchrony across oscillating receptors in the range of stimulus levels tested. Future comparative studies should investigate the effect of stimulus intensity on synchrony level across electroreceptors and whether the differences in sensitivity reported here relate to perceptual differences between species with spiking and oscillating receptors.

This study revealed conserved properties of the electrosensory system across mormyrids. In the three species studied, the knollenorgan electrosensory pathway is devoted to processing communication signals produced by other individuals, and midbrain evoked potentials were very similar in waveform and latency. These results suggest that some anatomical and physiological properties of the central electrosensory pathway are shared across all mormyrids. We also show differences between species with oscillating receptors and a small, undifferentiated EL relative to species from the two clades with spiking receptors and an enlarged ELa/ELp. Species with independent evolution of spiking receptors and an enlarged ELa/ELp (*P. microphthalmus* and *B. niger*) are more sensitive to short, high-frequency signals than species with oscillatory receptors and undifferentiated EL (*P. tenuicauda*). These results suggest parallel evolutionary changes in a neural circuit that allowed for a novel perceptual ability to detect signal waveform variation (*Carlson et al., 2011*). These results set the groundwork for future investigations into the anatomical and physiological basis of evolutionary change in sensory processing at cellular and circuit levels.

## Materials and methods

### Animals

All procedures for housing, handling, and testing animals were in compliance with the guidelines established by the National Institutes of Health and were approved by the Institutional Animal Care and Use Committee at Washington University in St. Louis. We obtained data from 8 *P. tenuicauda* (5 females, standard length [SL]: 6.9–7.6 cm, mass: 5.9–7.7 g; 3 males, SL: 6.5–9.6 cm, mass: 5.7–

16.4 g), 6 *P. microphthalmus* (4 females, SL: 6.9–7.2 cm, mass: 4.6–6.6 g; 2 males, SL: 7.0–7.1 cm, mass: 5.6–6.3 g), and 7 *B. niger* (3 females, SL: 6.6–8.4 cm, mass: 4.7–8.2 g; 4 males, SL: 7.1–8.5 cm, mass: 4.8–7.0 g). Readers are referred to *Baker et al. (2015)* for details about housing and handling individuals.

## Evoked potential recordings

The recording of evoked field potentials is a reliable and relatively simple technique that provides valuable insights into integrative synaptic processes within brain nuclei (*Einevoll et al., 2013*). The biophysical basis of local field potentials is disputed, and they may reflect neuronal spiking, synaptic activity, or both (*Anastassiou et al., 2015*; *Chizhov et al., 2015*; *Einevoll et al., 2013*; *Hall et al., 2014*; *Ray, 2015*). Regardless, it is clear that local field potentials represent summated electrical activity of neurons in the vicinity of the recording electrode. This technique has been widely used to study signal processing in ELa and ELp of clade-A species (*Amagai, 1998*; *Carlson, 2009*; *Lyons-Warren et al., 2012*, *2013a*; *Szabo et al., 1979*). Here, we followed previously described protocols to obtain evoked field potentials from the midbrain EL (*Carlson, 2009*; *Lyons-Warren et al., 2013b*). These evoked potential recordings do not reflect the diversity of single-neuron response properties in these brain regions, but instead a composite of responses across the population (*Carlson, 2009*; *George et al., 2011*; *Lyons-Warren et al., 2013a*).

Briefly, fish were first anesthetized with 300 mg/l of tricaine methanesulfonate (MS-222, Sigma–Aldrich, St. Louis, MO) and paralyzed with 100–150 µl of 3 mg/ml solution of gallamine triethiodide (Flaxedil, Sigma–Aldrich). We then transferred the fish to a recording chamber (20 × 12.5 × 45 cm) filled with fresh water, leaving a small region of the head above water level. We maintained general anesthesia for surgery by respirating the fish with an aerated solution of 100 mg/ml MS-222 through a pipette tip in the mouth. We applied Lidocaine (0.4%) as a local anesthetic at the surgery site. After removing the skin and securing a post to the skull, we removed part of the bone to expose the left ELa and ELp in *P. microphthalmus* and *B. niger*. In *P. tenuicauda*, the midbrain EL is not exposed as in the other two species. Therefore, we exposed the left EL by separating the optic tectum and the valvula cerebelli with two retractors made from borosilicate capillary glass. After surgery, we brought the fish out of anesthesia by switching to aerated fresh water respiration and monitored the fish's electromotor output with a pair of electrodes placed next to the fish's tail (*Carlson, 2002*). While flaxedil silences the EOD, a pair of external electrodes placed next to the tail can record EOD commands from spinal electromotor neurons. These EOD commands were amplified 1000x (Model 1700, A-M Systems) and sent to a window discriminator for time-stamping (SYS-121, World Precision Instruments).

We recorded evoked potentials using electrodes made of borosilicate capillary glass (o.d. = 1.0 mm, i.d. = 0.5 mm; A-M Systems, Model 626000) pulled on a Flaming/Brown micropipette puller (Sutter Instrument Company model P-97), broken to a tip diameter of 10–15 µm, and filled with 3M NaCl. Recordings of evoked potentials were obtained after the fish had completely recovered from anesthesia. Evoked potentials were amplified 1000x and band-pass filtered (0.01–5 kHz) with a differential AC amplifier (A-M systems, Model 1700), digitized at a rate of 97.6 kHz (Tucker Davis, Model RX 8), and saved using custom software in Matlab. We recorded responses to 10 repetitions of each type of stimulus. Evoked potentials are blocked during a short period of time after the EOD command (*Figure 2*). Therefore, in response to natural EODs and stimuli used to determine EL frequency tuning (see below), repetitions in which the fish emitted a command two to five ms before the stimulus were ignored and retaken. For each stimulus, we calculated a mean evoked potential by averaging the responses to the 10 repetitions.

## Stimuli and evoked potential analysis

We used three vertical electrodes on each side of the recording chamber (anodal to the left, cathodal to the right) to deliver transverse electrosensory stimuli with normal polarity (peak preceding trough). Digital stimuli were generated in Matlab (Mathworks), converted to analog and delivered with a Tucker-Davis RX8 signal processor, attenuated with a Tucker-Davis PA5 attenuator, and isolated from ground with an A-M Systems 2200 stimulus isolation unit.

To determine whether the central electrosensory system detects transient synchrony across oscillating receptors, we recorded evoked potentials in response to natural EODs. For each fish, one

natural EOD was selected at random from a library of 10 pre-recorded conspecific EODs. Natural EODs were delivered at a level of 31 mV/cm. We measured the latency to the first negative peak and the relative latency between the maximum and minimum values of the evoked potential.

To determine whether a corollary discharge of the electromotor command that initiates EOD production blocks sensory processing in the EL pathway, we delivered 0.5 ms bipolar square pulses (73.6 mV/cm) at latencies between 0 and 6 ms, in 0.5-ms steps, after the fish's EOD command. We measured the peak-to-peak amplitude of the mean evoked potential at each latency and normalized them to the maximum value for each fish. For *P. microphthalmus* and *B. niger*, we normalized mean evoked potentials in ELa and ELp separately. We used repeated-measures ANOVA in R (*R Core Team, 2014*) to investigate the effects of stimulus delay on the normalized peak-to-peak amplitude of the evoked potential.

We used single-cycle bipolar sine wave stimuli to measure the frequency tuning of EL. Sine wave stimuli were computer-generated using Matlab and had durations of 0.03, 0.05, 0.07, 0.1, 0.2, 0.3, 0.4, 0.5, 1, 2, and 5 ms. The peak-power frequency of these stimuli is slightly lower than the inverse of the pulse duration and ranges from 0.17 to 27.9 kHz (*Figure 4*). We used single-cycle bipolar sine wave pulses to be consistent with the previous study investigating peripheral frequency tuning (*Baker et al., 2015*). In that study, single-cycle pulse stimuli were necessary because it would be impossible to separate the stimulus artifact of longer tone stimuli from the oscillatory response of the oscillating receptors. Furthermore, single-cycle pulses are behaviorally relevant stimuli because they closely resemble the pulse-type communication signals of these species (*Figure 3A*).

Stimuli were delivered at intensities of 73.6, 23.4, and 7.4 mV/cm. We measured the peak-to-peak amplitude of the mean evoked potential for each duration and intensity, and normalized them to the maximum value for each fish. Mean evoked potentials were normalized separately for ELa and ELp in *P. microphthalmus* and *B. niger*. We used repeated-measures ANOVAs to investigate the effects of stimulus peak-power frequency and intensity on the normalized peak-to-peak amplitude of the evoked potential. We report $\eta^2$ in all of our models, which is the most common estimate of effect size for ANOVAs (*Levine and Hullett, 2002*).

All of the data and all of the custom-written Matlab codes used in this study are available at the Dryad Digital Repository (*Vélez and Carlson, 2016*).

## Acknowledgements

This research was supported by the National Science Foundation (IOS-1255396 to BAC).

## Additional information

### Funding

| Funder | Grant reference number | Author |
| --- | --- | --- |
| National Science Foundation | IOS-1255396 | Bruce A Carlson |

The funders had no role in study design, data collection and interpretation, or the decision to submit the work for publication.

### Author contributions

AV, BAC, Conception and design, Acquisition of data, Analysis and interpretation of data, Drafting or revising the article

### Author ORCIDs

Bruce A Carlson, http://orcid.org/0000-0002-2151-0443

### Ethics

Animal experimentation: All procedures for housing, handling, and testing animals were performed in strict accordance with the guidelines established by the National Institutes of Health and were approved by the Institutional Animal Care and Use Committee (Animal Welfare Assurance Number: #A-3381-01) at Washington University in St. Louis. The protocol was approved by the Animal Studies

Committee at Washington University in St. Louis (Approval Number: 20130265). Every effort was made to minimize pain and stress.

## Additional files

### Major datasets

The following dataset was generated:

| Author(s) | Year | Dataset title | Dataset URL | Database, license, and accessibility information |
|---|---|---|---|---|
| Vélez A,  Carlson BA | 2016 | Data from: Detection of transient synchrony across oscillating receptors by the central electrosensory system of mormyrid fish | http://dx.doi.org/10.5061/dryad.892f1 | Available at Dryad Digital Repository under a CC0 Public Domain Dedication |

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
