## [Decision Letter]

Thank you for submitting your article "Detection of transient synchrony across oscillating receptors by the central electrosensory system of mormyrid fish" for consideration by *eLife*. Your article has been favorably evaluated by Eve Marder (Senior editor) and three reviewers, one of whom, Ronald L Calabrese (Reviewer #1), is a member of our Board of Reviewing Editors. The following individuals involved in review of your submission have agreed to reveal their identity: Maurice J Chacron (Reviewer #2); John E Lewis (Reviewer #3).

The reviewers have discussed the reviews with one another and the Reviewing Editor has drafted this decision to help you prepare a revised submission.

Summary:

The authors present a very interesting advance to a previous publication in *eLife* in which they had demonstrated that spontaneously oscillating electroreceptors in weakly electric fish (Mormyridae) respond to electrosensory stimuli with a phase reset that results in transient synchrony across the receptor population. The advance here is that they extend this observation to the electrosensory lobe where electroreceptor sensory information is processed. They do this in the context of comparisons to other species that do not have oscillating, but spiking, receptors and are sensitive to electrosensory waveform variation. The analysis indicates that the oscillating receptor synchrony is processed faithfully in the EL of oscillating receptor fish, and the comparisons have interesting evolutionary implications: anatomical, physiological, and behavioral.

This advance should be a wide interest in the neuroethology community and also to other sensory physiologists and neuroscientists interested in the role of oscillations in brain networks (especially sensory networks) because it serves as an example of how synchrony can carry critical information.

Essential revisions:

1) The interpretation of the sinewave stimuli as in Figure 3 (also used in the first paper) is tricky. It appears that the stimulus is a single cycle sinewave "pulse", and so frequency is related to EOD pulse duration in a way. We ask the authors to clarify how the sinewave stimuli were obtained and delivered, perhaps by showing actual waveforms.

The Field Potential response is less sensitive to amplitude of this pulse in *P. tenuicauda* (oscillator species); FP amplitude also peaks at a much lower frequency (or duration) than the typical range of EOD pulse duration, suggesting that this species does not code EOD duration. It would enhance the impact of the advance to show how the level of synchrony changes over the stimulus amplitude range tested in Figure 3C (which shows relatively small changes in FP amplitude, and thus given a synchrony code, would suggest little change in synchrony). Synchrony in oscillating receptors should correlate with the relative lack of change in FP amplitude in *P. tenuicauda*. It should also correlate with changes in FP amplitude in species with spiking receptors. These data may be available from experiments already performed in the previous paper and this advance, and if so would we ask for these analyses rather than new experiments. If the expectation cannot be confirmed in existing data then the issue should be dealt with forthrightly in Discussion.

2) There is a general concern about the conclusions being over stated. The authors propose that their data "conclusively demonstrate a novel mechanism for sensory coding based on the detection of oscillatory synchrony among sensory receptors" that likely involves coincidence detection. The data presented are indeed consistent with this hypothesis, and the evoked potentials in EL provide a nice link to the previous study, where Baker et al. (2015) showed that *P. tenuicauda* respond behaviorally to conspecific EOD signals. At this point though, it is not clear that transient synchronization is 'necessary' for the evoked potentials recorded in EL. To show this, receptor synchronization must somehow be disrupted specifically, without changing the average activity over the population of receptors. This may be difficult to realize experimentally. A more systematic discussion, aimed at ruling out alternative hypotheses, could lend more support for the primary conclusion. There is also too much focus on coincidence detection in the nELL. The authors should either provide recordings from nELL neurons showing that they indeed act like coincidence detectors or tone this down to focus on their results. It seems that the authors propose that the coding mechanism is based on coincidence detection with different delays from input coming from both sides of the body in EL and not in nELL for species with spiking receptors. This needs to be contrasted with results obtained in species with oscillating receptors. Abstract should be appropriately edited.

3) The authors chose to record field potentials. The authors need to discuss the advantages and disadvantages of field potentials. Critically, it is not clear what causes these primarily in their recordings. Is it the spiking of post-synaptic neurons or spikes traveling down pre-synaptic axons? This question is important as the authors' data shows surprising similarity between field potentials recorded at i) and ii) in EL of *Petrocephalus tenuicaud*a and those recorded in ELa and ELp of *Petrocephalus microphthalmus*. Doesn't this undermine the argument that the differentiation of EL serves to underlie discrimination of EOD waveform in the latter species?

4) The authors need to clearly introduce the two groups (species with spiking and oscillatory receptors) and their differences, and very succinctly summarize the findings of their previous paper.

---

## [Author Response]

Essential revisions:

*1) The interpretation of the sinewave stimuli as in Figure 3 (also used in the first paper) is tricky. It appears that the stimulus is a single cycle sinewave "pulse", and so frequency is related to EOD pulse duration in a way. We ask the authors to clarify how the sinewave stimuli were obtained and delivered, perhaps by showing actual waveforms.*

The reviewers are correct in that the stimuli used are single-cycle sinewave pulses. The sine-wave pulses used in this experiment were computer-generated in Matlab. We specify this information in the Results and Methods of the revised manuscript in the first paragraph of the subsection “Midbrain frequency tuning reflects the intrinsic oscillation frequencies of oscillating receptors” and in the third paragraph of the subsection “Stimuli and Evoked Potential Analysis”, respectively. We also provide two graphical examples of the stimuli used and the evoked potentials obtained with those stimuli on Figure 4 of the revised manuscript.

In addition, the reviewers raise an important point regarding frequency content and pulse duration of our stimuli. The frequency content of a single-cycle sinewave pulse is inherently correlated with pulse duration. Therefore, our measurement of frequency sensitivity is, in a way, correlated with pulse-duration sensitivity. Importantly, however, frequency sensitivity is often interpreted in terms of the power spectrum of communication signals, and previous studies have demonstrated a clear correlation between the frequency tuning of knollenorgans and the peak power frequencies of species-specific EOD pulses, at least for clade A species (e.g. Hopkins, 1981; Baker et al., 2015). The communication signals of our three study species closely resemble single-cycle pulses (See Figure 3 of the revised manuscript) and therefore, a measurement of frequency sensitivity based on single-cycle pulses is biologically relevant. Furthermore, we used single-cycle sinewave pulses to be consistent with the previous study (Baker et al. 2015, *eLife*). In that study, single-cycle pulses were used instead of longer tone blips because the stimulus artifact generated by longer tones would be impossible to separate from the oscillatory response of the electroreceptors. We included this rationale in the Materials and methods section of the revised manuscript (subsection “Stimuli and Evoked Potential Analysis”, fourth paragraph).

The Field Potential response is less sensitive to amplitude of this pulse in P. tenuicauda (oscillator species); FP amplitude also peaks at a much lower frequency (or duration) than the typical range of EOD pulse duration, suggesting that this species does not code EOD duration. It would enhance the impact of the advance to show how the level of synchrony changes over the stimulus amplitude range tested in Figure 3 (which shows relatively small changes in FP amplitude, and thus given a synchrony code, would suggest little change in synchrony). Synchrony in oscillating receptors should correlate with the relative lack of change in FP amplitude in P. tenuicauda. It should also correlate with changes in FP amplitude in species with spiking receptors. These data may be available from experiments already performed in the previous paper and this advance, and if so would we ask for these analyses rather than new experiments. If the expectation cannot be confirmed in existing data then the issue should be dealt with forthrightly in Discussion.

The reviewers raise an interesting hypothesis about how the mormyrid electrosensory system may encode signal amplitude and they propose a neat experiment to test it. Unfortunately, we do not have data related to the proposed experiment, and comparing the results from Baker et al. (2015, *eLife*) with ours is not straightforward in addressing the question of amplitude coding. In the previous study, single electroreceptors were stimulated with constant-current pulses in nA. In our study, we used global electrosensory stimuli to stimulate all receptors at constant electric fields measured in mV/cm at the position of the fish in the test tank. It is not clear how a given electric field intensity will relate to current magnitude at a given receptor pore. Therefore, it is not possible to accurately relate the intensity of the electric field experienced by the fish in our experiment with the constant current used to stimulate individual electroreceptors in the previous study. For this reason, we are unable to include these analyses in the revised manuscript. Nevertheless, we discuss this important issue in the fifth paragraph of the Discussion of the revised manuscript.

*2) There is a general concern about the conclusions being over stated. The authors propose that their data "conclusively demonstrate a novel mechanism for sensory coding based on the detection of oscillatory synchrony among sensory receptors" that likely involves coincidence detection. The data presented are indeed consistent with this hypothesis, and the evoked potentials in EL provide a nice link to the previous study, where Baker et al. (2015) showed that P. tenuicauda respond behaviorally to conspecific EOD signals. At this point though, it is not clear that transient synchronization is 'necessary' for the evoked potentials recorded in EL. To show this, receptor synchronization must somehow be disrupted specifically, without changing the average activity over the population of receptors. This may be difficult to realize experimentally. A more systematic discussion, aimed at ruling out alternative hypotheses, could lend more support for the primary conclusion.*

We appreciate the reviewers bringing up this concern, and we agree with them that more evidence is necessary to conclusively demonstrate this mechanism. Therefore, we toned down this conclusion and state it as further evidence in support of a model for sensory coding based on the detection of oscillatory synchrony. We rephrased the Abstract and Discussion (first paragraph) in the revised manuscript, taking into consideration this comment. In addition, we expanded our Discussion to include alternative hypotheses about the mechanisms underlying our results (second paragraph). Since electrosensory stimulation elicits an increase in the amplitude of oscillations and a phase reset that causes transient oscillatory synchrony, we discuss the possibility that the electrosensory system detects a change in oscillatory amplitude. Given the available evidence and temporal precision of the responses, we suggest that the underlying mechanism for sensory coding is more likely the detection of transient oscillatory synchrony than the detection of a change in oscillatory amplitude, but the increase in amplitude likely makes detecting this synchrony an easier task.

*There is also too much focus on coincidence detection in the nELL. The authors should either provide recordings from nELL neurons showing that they indeed act like coincidence detectors or tone this down to focus on their results.*

To appease the reviewers’ comments, we significantly shortened (by 20% ) the section devoted to nELL as the location of coincidence detection in our Discussion (third paragraph in revised manuscript). In addition, we rephrased it to make it clear that this is a hypothesis. However, we decided not to remove it because we consider it to be a valid hypothesis backed up by previous research, it is likely to be of relevance to a large proportion of the expected readers of our manuscript, and it sets the groundwork for future comparative physiological and anatomical studies.

It seems that the authors propose that the coding mechanism is based on coincidence detection with different delays from input coming from both sides of the body in EL and not in nELL for species with spiking receptors. This needs to be contrasted with results obtained in species with oscillating receptors. Abstract should be appropriately edited.

We apologize for the confusion generated around coincidence detection in nELL. The available evidence suggests that species with spiking receptors perform coincidence detection in the nELL so that nELL neurons only spike in response to synchronous input from multiple receptors with adjacent receptive fields, as we propose for species with oscillatory receptors (Bell and Grant 1989). Studies of ELa in species with spiking receptors instead suggest that anti-coincidence detection of inputs from widely separated receptive fields is used in signal waveform analysis (Friedman and Hopkins 1998; Lyons-Warren et al. 2013). We included in the Introduction of the revised manuscript a brief description of what is currently known about information processing in the ascending knollenorgan pathway (fourth paragraph). In this paragraph, we explain the projections from the electroreceptors to the hindbrain nELL (ipsilateral), from the hindbrain to ELa (bilateral), and from ELa to ELp (ipsilateral). In addition, we briefly explain how cells in ELa perform EOD waveform analysis. With this explanation in the Introduction, we hope to have cleared up any confusion.

We also rephrased the Abstract to account for the toned-down discussion about nELL.

3) The authors chose to record field potentials. The authors need to discuss the advantages and disadvantages of field potentials. Critically, it is not clear what causes these primarily in their recordings. Is it the spiking of post-synaptic neurons or spikes traveling down pre-synaptic axons? This question is important as the authors' data shows surprising similarity between field potentials recorded at i) and ii) in EL of Petrocephalus tenuicauda and those recorded in ELa and ELp of Petrocephalus microphthalmus. Doesn't this undermine the argument that the differentiation of EL serves to underlie discrimination of EOD waveform in the latter species?

Evoked field potentials have been widely used to study signal processing in mormyrids and in other systems. Evoked potentials reflect summated activity across many neurons; therefore, one caveat of this technique is that it does not reflect the diversity of properties observed at the level of single neurons. However, evoked potentials are a reliable and relatively simple technique that provides valuable insight into signal processing and integrative synaptic processes in these nuclei, and generate data that can be compared to a large number of studies that have also used this technique. We have included this information together with a brief discussion of what evoked potentials represent in the Materials and methods section of the revised manuscript (subsection “Evoked Potential Recordings”, first paragraph).

In addition, we rephrased the last paragraph of our Discussion to emphasize the similarities and the differences that we found between species with oscillating receptors and small EL as compared to species with spiking receptors and enlarged, subdivided ELa/ELp. The similarities suggest that some properties of the electrosensory system are conserved across all mormyrids. The differences suggest parallel evolutionary change in a neural circuit that allowed for a novel perceptual ability in two groups of mormyrids. These similarities and differences do not undermine the argument that the differentiation of EL serves to underlie discrimination of EOD waveform, because the evoked potentials do not capture what is happening on a single-neuron level, only a population level. Further, we did not address the sensitivity of single neurons to variation in EOD waveform, but of evoked potentials to variation in pulse duration/frequency content (analyzing sensitivity to pulse waveform per se requires temporal manipulations of the waveform that do not alter the frequency content, as in Carlson et al., 2011). Among the differences that we emphasize between these two groups of species is the ability to process short, high-frequency signals. Rather than undermining this argument, our results set the stage for future studies investigating the differences and similarities at cellular and circuit levels that allow species with ELa/ELp (but not species with small EL) to detect variation in EOD waveform.

*4) The authors need to clearly introduce the two groups (species with spiking and oscillatory receptors) and their differences, and very succinctly summarize the findings of their previous paper.*

We rephrased our entire Introduction to introduce and clarify the differences between groups of fish, what we currently know about signal processing in mormyrids, and to emphasize the findings of the previous study and how we are building upon those findings.